# The Genetic Diversity of Mink (*Neovison vison*) Populations in China

**DOI:** 10.3390/ani13091497

**Published:** 2023-04-27

**Authors:** Tietao Zhang, Hu Li, Peter Foged Larsen, Hengxing Ba, Hongyu Shi, Haihua Zhang, Zongyue Liu

**Affiliations:** 1Jilin Provincial Key Laboratory for Molecular Biology of Special Economic Animals, Key Laboratory of Special Economic Animal Genetic Breeding and Reproduction, Ministry of Agriculture, Institute of Special Economic Animal and Plant Sciences, The Chinese Academy of Agricultural Sciences, Changchun 130112, China; 2Colleges of Animal Science, Hebei Normal University of Science and Technology, Qinhuangdao 066004, China; 3Institute of Antler Science and Product Technology, Changchun Sci-Tech University, Changchun 130112, China

**Keywords:** mink, genetic diversity, population structure, selective sweeps, dd-RAD

## Abstract

**Simple Summary:**

Mink (*Neovison vison*) is well-known as one of the most important sources of fur and is used as feed across the North zone of China. Using double-digest restriction site-associated DNA sequencing, this study evaluated the genetic diversity and population structure of a five color of mink populations in China (ddRAD-seq). The black mink population was found to be genetically differentiated from other color type populations, whereas a clustering of the red mink and black mink populations was observed; further, other color populations clustered separately. Gene Ontology and Kyoto Encyclopedia of Genes and Genomes analysis showed that the genes with a selection signature were enriched in the melanogenesis pathway. The results of our study provide basic information on mink diversity, and theoretical concepts for the conservation and exploitation of mink breeds in China.

**Abstract:**

The American mink (*Neovison vison*) is a semiaquatic species of Mustelid native to North America that is now widespread in China. However, the knowledge of genetic diversity of mink in China is still limited. In this study, we investigated the genetic diversity and identified significant single nucleotide polymorphisms (SNPs) in mink populations of five different color types in three different mink farms in China. Using double-digest restriction site-associated DNA sequencing, we identified a total of 1.3 million SNPs. After filtering the SNPs, phylogenetic tree, Fst, principal component, and population structure analyses were performed. The results demonstrated that red mink and black mink grouped, with separate clustering of all other color types. The population divergence index (Fst) study confirmed that different mink populations were distinct (K = 4). Two populations with different coat colors were subjected to the selection signature analysis, and 2300 genes were found to have a clear selection signature. The genes with a selection signature were subjected to Gene Ontology (GO) categorization and Kyoto Encyclopedia of Genes and Genomes (KEGG) enrichment analysis, the results revealed that the genes with a selection signature were enriched in the melanogenesis pathway. These study’s findings have set the stage for improved breeding and conservation of genetic resources in real-world practical mink farming.

## 1. Introduction

The farm mink (*Neovison vison*) is a semi-aquatic carnivore that traditionally originated from the Northern Hemisphere [1]. Due to their importance in the fur industry, these animals and their skins were transported from Western Europe to the Far East, especially China in the late 20th century [2]. According to the data for 2021, approximately 20 million mink skins are produced globally; China contributed 25% of the total mink skin produced worldwide [3]. In China, almost all mink farms are located in the Northeast zone and around the Bohai Sea. Although China lacks a local mink, many breeds of mink were imported from Russia, North America, and North Europe [2]. In an aim to avoid decrease in the number of purebred mink performances, the crossing method was used to foster the new type in most of the mink farms. This practice has led to the official recognition of 13 mink breeds, of which seven and six are domestically fostered and introduced breeds, respectively [4].

Several previous studies have specifically investigated the genetic diversity of the mink population or color type using microsatellites and mitochondrial genes, for instance, mitochondrial cytochrome b and 12S rRNA genes in seven populations [5], 20 microsatellites in eight mink populations [6], 30 inter-simple sequence repeat (ISSR) markers in five color types [7] and 14 random amplified microsatellite polymorphism (RAMP) markers in three color types [6]. Furthermore, the genetic structure of farm and feral American mink was previously studied using 11 microsatellite markers in Japanese mink [8], 12 microsatellites and sequencing of mitochondrial cytochrome b gene in farm and feral American mink populations [9], and mitochondrial DNA sequences along with 11 microsatellite markers in Chilean mink [10].

Single nucleotide polymorphisms (SNPs) are ideal molecular markers for genetic mapping and population diversity assessment [11], because of their high density and relatively uniform distribution throughout the genome. SNPs are extensively used in livestock species such as cattle [12], sheep [13], goats [14],and pigs [15]. The first draft genome of mink was published by Cai et al. in 2017 [16], whereas *Younes Miar* [17] assembled the accurate whole-genome sequence of mink using next-generation sequencing technology (PacBio, HiSeq, and Hi-C), opening several opportunities for genomic selection in mink. Although SNPs genotyping array has not been developed for mink as yet, SNP genotyping can be obtained using methods, such as sequencing (GBS), restriction site-associated DNA sequencing (RAD-seq), whole genome sequencing (WGS), and genotyping by targeted sequencing (GBTS). Thirstrup et al. [18] identified 380 SNPs at the genome level in the Danish mink population by RAD-seq. Furthermore, 13,321 SNPs were extracted using GBS in American mink [19]. Scanning of the genome revealed the putative genomic regions in American mink responsible for the response during Aleutian disease, which is caused by virus infection [20]. Inspite of these advances, the basic genetic information on the mink population in China is still lacking.

This study aimed to discover genome-wide SNPs in different mink populations (color type) in China and to determine the genetic diversity and structure, and the important role of genes in different mink populations in China.

## 2. Materials and Methods

### 2.1. Sample Collection and DNA Extraction

A total of 97 unrelated minks were obtained from three different farms in China and included 50 white minks (WM), 20 black minks (BLM), 10 sliverblue minks (SM), 10 brown minks (BM), and 7 red minks (RM). WM, SM, and BM were from Huilongwan breeding mink farm in Yichun City; BLM was from Zhuojia mink farm in Jilin City; and RM was from Shandong mink farm in Weifang City. Blood samples were obtained from the paws of minks and stored at −20 °C. Genomic DNA was extracted from these whole blood samples using Easy Pure Blood Genomic DNA kit according to the manufacturer’s instructions, and each DNA sample was evaluated using gel electrophoresis and stored at −20 °C. Animal management and sampling procedures were performed following the standards of the Principle of Laboratory Animal Care and the guidelines prescribed by the Animal Research Committee of the Institute of Special Animals and Plants Sciences, Chinese Academy of Agricultural Sciences (Protocol code no. ISAPSAEC-2022-62M).

### 2.2. Library Construction and Variant Calling

Double digest RAD sequencing (ddRAD-seq) libraries were constructed following a modified protocol [21]. Genomic DNA was digested by PstI and MspI enzymes, and resulting DNA fragments were amplified and subsequently purified by VAHTSTM DNA Clean Beads. After separation on a 2% agarose gel, DNA in the range of 220–450 bp (with indices and adaptors) was excised using a gel extraction kit (Qiagen, Hilden, Germany). Finally, the samples were sequenced by the Illumina NovaSeq platform at Shanghai Personal Biotechnology Co., Ltd., Shanghai, China.

Each sample was assessed in terms of the quality of sequencing reads using the Fast QC software (v0.11.7) [22]; Low-quality reads were trimmed l, and overlapping reads (at least 11 nucleotides, per default) were merged (collapsed). The resulting high-quality reads were aligned to the latest reference genome of the standard black mink, which accessed on 16 December 2022 (https://www.ncbi.nlm.nih.gov/assembly/GCF_020171115.1/) using BWA-mem v 0.7.12 [23]. High confidence SNPs were called by GATK (V3.8). Variants were filtered by GATK to using the criteria a quality by depth (QD) ˂ 2.0, mapping quality (MQ) < 40.0, Fisher strand (FS) > 60.0, mapping quality rank-sum test < −12.5, and read position rank sum test < −8.0. Furthermore, all SNPs with a minor allele frequency (MAF) < 0.05, call rate < 0.90 and those deviating from the Hardy–Weinberg equilibrium (*p* < 10^−6^) were filtered out. In addition, individuals with >0.15 missing genotype were discarded from the dataset using VCFtools 35. Minor allele frequency was computed for all SNPs and the proportion of SNPs was determined for MAF ranges of <0.05, 0.05 to <0.1, 0.1 to <0.2, 0.2 to <0.3, 0.3 to <0.4 and 0.4 to ≤0.5. Functional annotation of the SNPs and indels was performed using the ANNOVAR2017 software (2017-07-16) [24].

### 2.3. Genetics Analyses

A total of 1,396,257 SNPs were used to calculate the phylogenetic distance among mink populations. The phylogenetic tree was constructed using FastTree (version 2.1.11) software to determine the evolutionary relationship among mink populations. Principal component analysis (PCA) was performed using the GCTA software (version 1.26.0). Plink was used to analyze the genetic structure of the mink populations based on different color type. The number of genetic clusters was predefined from K = 2 to 10 and repeated each K-value three times. Admixture software (Version 1.3.0) was used to calculate ΔK. Observed heterozygosity (Obs Het), observed homozygosity(Obs Hom), expected heterozygosity (Exp Het), expected homozygosity(Exp Hom), nucleotide diversity (π) and inbreeding coefficients (Fis) were calculated using the populations’ program in STACKS. The Fst values were calculated for each SNP of all pairwise subgroups using variant call format (VCF) tools. We studied the genetic structure of different mink populations using phylogenetic tree construction, PCA, and population structure analysis, whereas the genetic analysis of population structure was carried out using ADMIXTURE software (Version 1.3.0).

### 2.4. Identification of Selective Sweeps

Candidate genes in the sweep regions of different mink populations were analyzed using Gene Ontology (GO) and Kyoto Encyclopedia of Genes and Genomes (KEGG) databases. Top GO software (Version: 2.38.1) was used to perform the GO enrichment analysis and the pathway information is fetched from KEGG website. phyper is used to do the hypergeometric tests. Calculated *p*-values were corrected for the false discovery rate (FDR) while applying an FDR threshold of ≤0.05. Pathways meeting the criteria were considered significantly enriched pathways.

## 3. Results

### 3.1. ddRAD-Seq and Data Filtering

Before quality filtering, ddRAD-seq generated 390.08G of raw data for 97 normally sequenced individuals, with an average of 4.02 Gb per sample and a range of 2.31 to 6.23 Gb. 362.63 Gb of clean data (2.16 Gb to 5.21 Gb for each sample, with an average of 3.74 Gb) were retained after the sequence data underwent quality filtering. Each sample received an average of 26.71 million clean reads that were retained. The number of reads on the alignment was mostly above 93.21%, the percentage of high-quality clean reads was above 94.52%, and the average effective mapping rate was above 98.86%. Hence, our sequencing data were of a high quality (Q20 > 96%, Q30 > 91%), with a stable GC content that ranged from 49.89 to 54.95% (Appendix A).

### 3.2. Variation Calling

A total of 1,396,257 high-quality SNPs were identified after SNP calling on 97 samples, with sequencing depths ranging from 19.05 to 20.95 (Table 1). Each animal had the same number of SNPs, which ranged from 9,211,330 in the BM population to 1,4738,128 in the red mink, but the number of SNPs in each individual was the same, and the number of identified homozygous SNPs varied from 203,639 to 234,273 and from 74,386 to 88,124 (Table 2). Furthermore, 36,286 SNPs were detected in exon regions, which included 19,444 nonsynonymous, 16,420 synonymous, 382 stop-gain and 27 stop-loss variants (Table 3). The transition to transversion (Ti/Tv) ratio of SNPs was 2.45. A total of 205,733 InDels identified; insertions and deletions ranged from 1−44 bp and 1−69 bp, respectively. Of these, a substantial proportion (91.65%) of indels were relatively small (1–6 bp), with only 2.14% (n = 4545) greater than 20 bp (Figure 1, Appendix A).

### 3.3. Genetic Diversity

We detected deficits in all Obs values in all mink populations and the Obs values were compared with the Exp values. RM showed the lowest Exp Het, whereas the highest Obs Het was observed in BLM, indicating the occurrence of self-crossing in the populations. The intra-population inbreeding coefficient of Fis ranged from 0.2441 to 0.4402. We estimated the genome-wide nucleotide diversity using the SNP data. π values, which ranged from 0.2099 to 0.3503, demonstrated that the nucleotide diversity of BLM was the highest. The average Obs Het, Exp Het, π, and Fis were 0.1226, 0.1948, 0.2272, and 0.2308, respectively (Table 4). The estimated genetic distances according to the Fst metric varied widely between 0.0833 and 0.1657 among the populations (Table 5). The highest and lowest genetic distances were observed between RM and SM and BM and WM, respectively.

### 3.4. Genetic Structure

Similar patterns were shown by genetic analysis of the phylogenetic tree and PCA. The findings indicated that SM, BM and WM were grouped independently. Some minks were congregated, with RM minks clustered with BLM minks, while others distinguished themselves and created a new group (Figure 2A,B). The ideal value of K was 3 based on each k’s cross-validation error (Figure 2C). However, the clusters were not produced from BLM, BM, or SM. When K was adjusted to 2, RM was separated from other populations, and the latter WM was further separated when K was set to 3. Several mink populations were divided at K = 4.

### 3.5. Selection Signatures and Enrichment

Fst and π, which were selected in the top 5% of regions, were used in selection sweeps. Five mink populations’ comparisons of 2300 genes with a selection signature revealed 10 highly enriched terms (*p* < 0.05), including three functional terms for biological processes (BP), four for molecular functions (MF), and three for cellular components (CC). For BP terms, the cysteinyl-tRNA aminoacylation, columnar cuboidal epithelial cell growth, and sulfur amino acid catabolic process were determined to have the strongest GO connections. For the MF keywords, thiamine binding (GO: 0006772), flavin-linked sulfhydryl oxidase activity, cysteine-type peptidase activity, and melatonin receptor activity (GO: 0008502). Contrarily, CC keywords were connected to the melanosome (GO: 0042470), pigment granule (GO: 0048770), and photoreceptor outer segment. Some of the most important genes were found to be abundant in the melanogenesis pathway, cysteine and methionine metabolism, Wnt signaling route, MAPK signaling system, and phototransduction in the KEGG enrichment analysis of the genes with a selection signature. The development of melatonin and the hair structure was also linked to 12 genes, including Wnt11, Wnt5B, PSMB3, CCND1, MTNR1B, EGF, DN-EGF, WD, WIPI1, IGFII, IGFBP7, BMP6, and TGF-β (Figure 3, Appendix A).

## 4. Discussion

ddRAD-seq and other related approaches can provide a picture of the distribution of SNPs throughout the whole genome, they have been widely used in genetic and population structure analyses [25]. In the present study, a total of 1.39 M SNPs were identified from 97 animals belonging to five different color populations, enabling us to predict the relationships among these populations of animals. This number of SNPs identified was significantly higher than that reported previously, for instance, 52,714 SNPs of the LD pattern across the American mink genome by GBS [26] and 34,816 SNPs with body size and pelt length in American mink by GBS [16]. This should be attributed to the use of different restriction enzymes that reduce the complexity of the genome for sequencing. Furthermore, 98.86% of clean reads were mapped to the mink reference genome, with subsequent SNP calling. These data are consistent with the population genomic research (on average, 98.24%) performed using whole genome sequencing [20] in American minks. This indicated that the identified SNPs can be used in downstream analysis.

The Fst varied between 0.0839 and 0.2241 for various color types, which is within the range of another study conducted at 12 locations in southern Chile (Fst varied between 0.017 and 0.364) [10]. According to the findings of this research, Thirstrup et al. [18] revealed modest genetic divergence (Fst) across mink color types (0.076) and between mink from two distinct geographic origins (0.087), with the highest Fst (0.14) reported between Pastel and Stardust color-types in American mink.

From the PCA, phylogenetic tree map, and structure analyses, most animals of RM and BM populations clustered together, whereas other populations (SM, BM, and WM) clustered separately. It indicates that the genetic relationship of between BM and RM populations was relatively close. The standard type of American mink in nature was more brownish than the present type of standard black and standard brown, and standard black was the result of a selection of dark colors among the original dark brown wild minks [27]. The WM of this study was Albino, also called pink-eyed white mink, which was a recessive mutant of standard mink. The first mutant of the regular mink, the Sliverblue mink, first appeared in Wisconsin, USA, in 1931. Farm samples displayed a distinct pattern of genetic subdivision in a prior study, and White, Pearl, Silver, and Sapphire were easily distinguished as distinct clusters with high odds of population membership [18]. The majority of the RM in China was imported from Russia in the 1950s, while the black mink in our study was imported from North America in the 1990s when the American mink was first introduced as a furbearer to the eastern and southern former Soviet Union [28]. Russian mink were huge with long, coarse guard hair, brown fur, and subpar reddish wool.

To enhance the quality of the Red mink’s fur, Black minks, which were little but had silky, short guard hair and dense wool, were utilized [29]. Our findings were in line with a previously published study of several color types of mink, in which structural analysis revealed that Mahogany looked to represent an admixture between Black and Brown mink.Since the color-phase breeding approach is used, genetic selection or genetic drift may have varied effects on mink depending on their line and farm [30]. Due to the frequent occurrence of the drift in cconfined populations of animals created by tiny, isolated populations, American mink are particularly vulnerable to this genetic mechanism [31].

Variation in coat color is thought to be a marker of domestication and a distinct trait of a few breeds [32].. In contrast, coat color and fur quality were essential features in mink breeding generally and in the mink industry specifically because of their effects on the eventual economic worth of fur [33].

Over a century of the occurrence of new mutations and selection on the initial wild brown and black phenotype resulted in 35 mutations and more than 100 forms from their combination (Table 6). Mammals’ various coat colors are initially related to the sort of melanin that melanocytes generate. In our work, GO analysis was enriched to demonstrate significant population variation in melanosome, pigment granule, melatonin receptor function, phototransduction, and photoreceptor outer segment. Coat color is determined by the amounts and types of melanosomes produced and released by melanocytes resident in the skin [34]. The principal role of the melanocyte is to produce the pigment granule within melanosomes and to transfer these organelles to keratinocytes [35]. Melatonin, a significant indoleamine neuromodulator implicated in circadian rhythms and sleep patterns, regulates diverse rhythmic functions by activating its high-affinity G-protein-coupled receptors [35]. In the adult rat retina, the melatonin receptor MT 1 immunocytochemically locates in the IPL, OPL, and ipRGCs, while MT 1 receptor immunoreactivity is found in many mouse retinal neurons, including rod and cone photoreceptors and ipRGCs [36]. In our research, the melanogenesis pathway, Wnt signaling pathway, MAPK signaling pathway, Cysteine, and methionine metabolism, and phototransduction were enriched by 11 genes, including Wnt 11, Wnt5B, PSMB3, CCND1, TGFβ, MTNR1B, EGF, DN-EGF, WD, WIPI1, and IGFII. In the International Federation of Pigment Cell Societies(IFPCS), 661 genes related to coat color have been published so far [37]. The research on fur development mainly focuses on the molecular mechanism of hair follicle development. Most of these key signaling molecules of hair follicles belong to Wnt (Wingless-related), Shh (Sonic Hedgehog), BMP (Bone morphogenetic protein), FGFs (Fibroblast growth factors), TGF(Transforming growth factor), and Notch signaling pathway [38]. Hsiang Ho [39] found that WD repeats domain, phosphoinositide-interacting 1 (WIPI1) coordinates melanogenic gene transcription and melanosome formation. Fibroblasts act on melanocytes directly and indirectly through neighboring cells by secreting a large number of cytokines, proteins, and growth factors (KGF, bFGF, TGF-β) which bind to receptors and modulate intracellular signaling cascades (MAPK/ERK, Wnt/β-catenin) related to melanocyte functions [40]. The Wnt pathway was significant in fur quality, which is the biological pathway considered to be the key regulator of hair follicle morphogenesis [41]. The canonical Wnt pathway has been reported for its important role in skin pigmentation and melanogenesis in chickens [18]. In our research, we found Wnt 11, Wnt5B, PSMB3, and CCND1 in the Wnt signaling pathway. Many cell growth factors have been reported to be regulators of hair follicle growth and development, including hepatocyte growth factor (HGF), insulin-like growth factor-1 (IGF-1), and epidermal growth factor (EGF). Wu [42] discovered that BMP6 and Wnt10b cooperate to control the activation of HFSCs and the transition of hair follicles from telogen to anagen. Consequently, our findings show that a range of genes, including WD, WIPI1, TGF, Wnt, and EGF, can influence melanin deposition in minks in different ways to change the color of the mink coat. In conclusion, selection signature analysis was used to investigate the genetic analyses and genomic resources related to the coat color of Chinese mink breeds. Our findings revealed 1.39 million SNPs in five color populations, which may be used to create SNP arrays for upcoming genomics initiatives in the Chinese mink population.

## 5. Conclusions

In this study, we explored the genetic diversity, and genetic organization of different mink in China using the ddRAD-seq technology to discover SNPs at the genome level. Also, we discovered the potential genes that were associated with melanogenesis. The findings of our study offer important data for the preservation and use of Chinese mink breeds, as well as a clear direction for upcoming breeding efforts.

## Figures and Tables

**Figure 1 animals-13-01497-f001:**
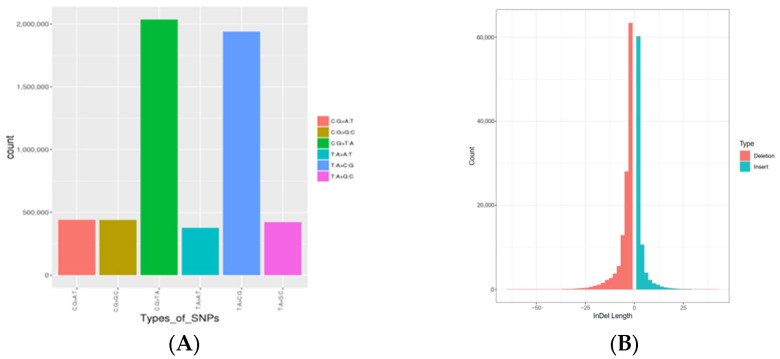
(**A**) Mutation spectrum analysis. (**B**) Diagram showing the distribution of indel length.

**Figure 2 animals-13-01497-f002:**
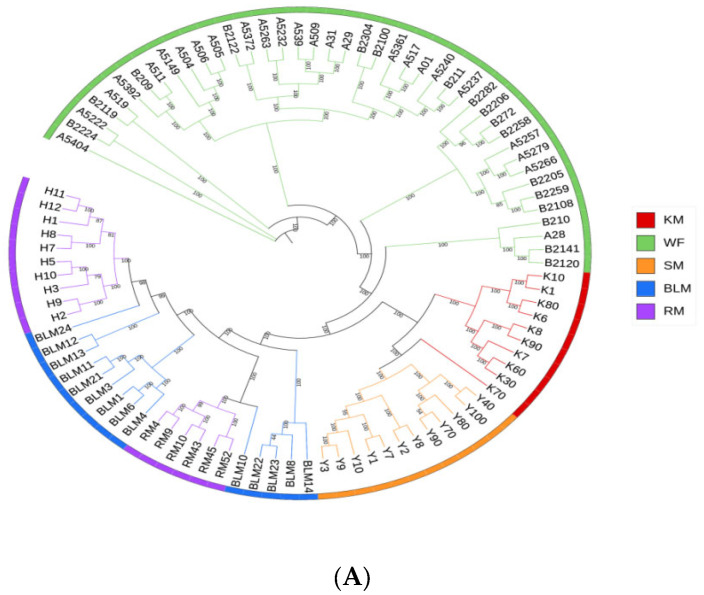
Genetic structure of the different populations. (**A**) Phylogenetic tree based on SNP. (**B**) PCA analysis. (**C**) Population genetic structure. Note: WF: white mink; BLM: black mink; SM: silverblue mink; KM: brown mink; RM: red mink.

**Figure 3 animals-13-01497-f003:**
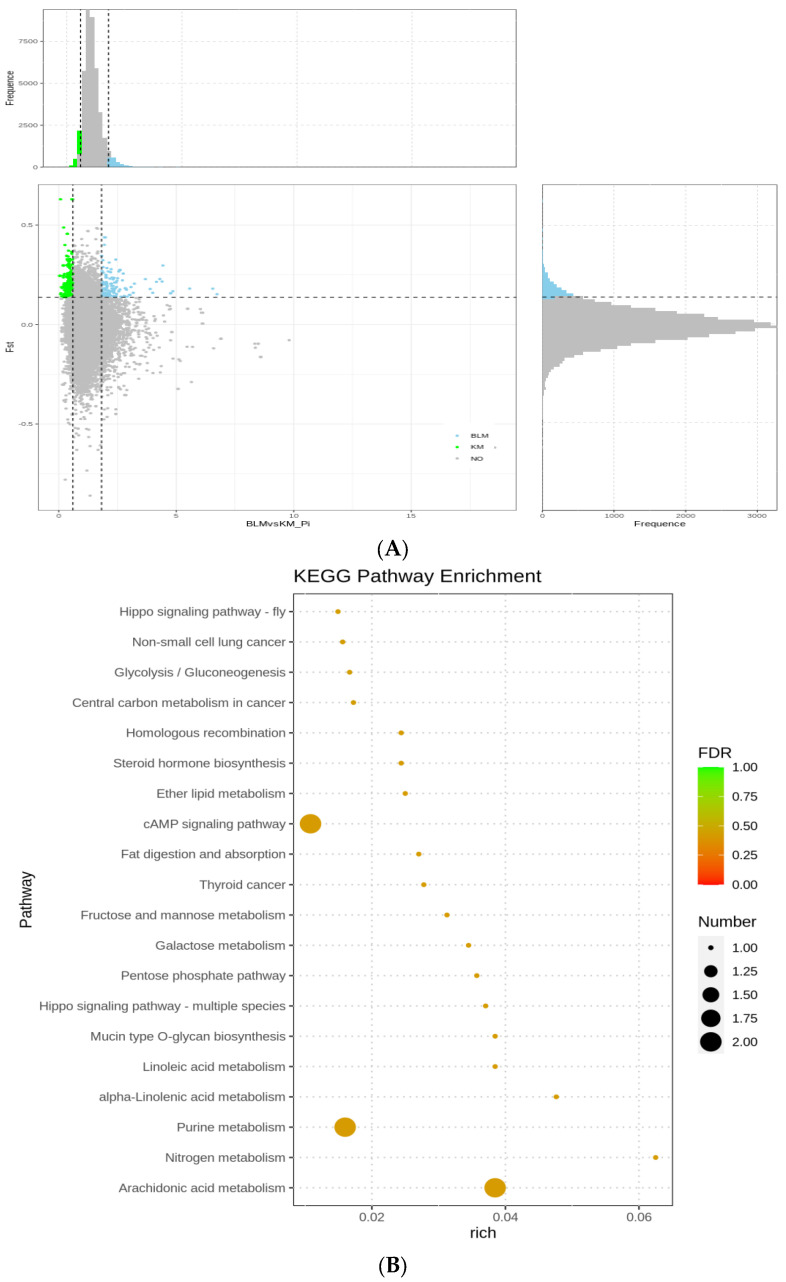
GO classification and KEGG enrichment of the selected genes of SM (BLM_versus_SM). (**A**) BLM was the control group and SM was the selection group; 307 selected genes were obtained. (**B**) KEGG enrichment of the selected genes of SM. (**C**) GO classification of the selected genes of SM.

**Table 1 animals-13-01497-t001:** Mapping of RAD sequencing reads to the reference genome of mink (GCA_020171115.1).

Population	Average Sequencing Depth	Coverage	Coverage at Least 4X	Coverage at Least 10X	Coverage at Least 20X
WM	1.44	7.44%	4.79%	3.32%	2.21%
BLM	1.66	8.56%	5.59%	3.91%	2.56%
BM	1.56	7.88%	5.06%	3.56%	2.38%
RM	1.54	8.08%	5.24%	3.69%	2.43%
SM	1.49	7.11%	4.48%	3.13%	2.22%

Note: WM: white mink; BLM: black mink; SM: sliverblue mink; BM: brown mink; RM: red mink.

**Table 2 animals-13-01497-t002:** SNP information of each population.

Population	HOM_REF	HET	UNKNOWN	HOM_ALT	SUM
WM	200,812	49,808	594,182	76,330	38,687,586
BLM	225,361	67,709	542,191	85,870	12,895,862
BM	218,001	50,891	574,606	77,633	92,11,330
RM	234,273	65,438	533,296	88,124	14,738,128
SM	203,639	40,290	602,817	74,386	11,053,596
Total	1,082,086	274,136	2,847,092	402,343	86,586,502

Note: WM: white mink; BLM: black mink; SM: silverblue mink; BM: brown mink; RM: red mink; HOM_REF: the sample is homozygous reference; HET: the sample is heterozygous; UNKNOWN: the sample contains missing genotype; HOM_ALT: the sample is homozygous alternate.

**Table 3 animals-13-01497-t003:** Mutations and their location in the genome.

Type	All	
	Number	Percentage
exonic	36,286	2.6
synonymous SNV	19,444	1.39
nonsynonymous SNV	16,420	1.18
stop-gain	382	0.03
stop-loss	27	0
unknown	13	0
splicing	261	0.02
ncRNA total	0	0
ncRNA exonic	0	0
ncRNA splicing	0	0
ncRNA exonic; splicing	0	0
ncRNA intronic	0	0
intronic	491,190	35.18
intergenic	829,525	59.41
UTR5	1990	0.14
UTR3	5321	0.38
UTR5; UTR3	4	0
upstream	15,222	1.09
downstream	16,057	1.15
upstream; downstream	401	0.03
Total	1,396,257	100

Note: SNV: single-nucleotide variant; UTR: untranslated region.

**Table 4 animals-13-01497-t004:** Comparison of population genetic parameters in different mink groups.

Population	Obs Het	Obs Hom	Exp Het	Exp Hom	π	Fis
BLM	0.1317	0.7683	0.2725	0.7275	0.3503	0.2641
BM	0.105	0.895	0.1971	0.8029	0.2209	0.2517
RM	0.1167	0.8833	0.1928	0.8072	0.2099	0.2441
SM	0.1006	0.8994	0.1961	0.8039	0.2261	0.2786
WM	0.1089	0.8911	0.2552	0.7448	0.2731	0.4402

Note: WM: white mink; BLM: black mink; SM: sliverblue mink; BM: brown mink; RM: red mink; π: Nucleotide diversity; Obs Het: Observed heterozygosity; Obs Hom: Observed homozygosity; Exp Het: Expected heterozygosity; Exp Hom: Expected Homozygosity; Fis: inbreeding coefficient.

**Table 5 animals-13-01497-t005:** Estimation of Fst between different mink populations.

	BM	RM	SM	WM
BLM	0.134487	0.148431	0.148311	0.110624
BM		0.154566	0.136017	0.083326
RM			0.16579	0.11867
SM				0.096049

Note: WM: white mink; BLM: black mink; SM: sliverblue mink; BM: brown mink; RM: red mink; Fst; the population differentiation index.

**Table 6 animals-13-01497-t006:** Major coat color phenotypes and associated genes in mink.

Coat Color Name	Symbol	Gene	Reference
STANDRAD			
White types			[43,44]
Albino	cc	MITF	
Hedlund	hh	TYR	
Grey types			
Sliverblue	pp	MLPH	
Steelblue	psps		[45,46]
Aleutian	aa	LYST	
Brown types			
Palomino	kk	TYRP1	[47,48]
Moyle buff	mm	RAB38	
Patterned			
Goofus	oo		
Others			
Shadow	SRSR	KIT	[49]
Black crystal	Cr Cr	COPA	[50]

## Data Availability

Data will be available upon request.

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
