# Peer review of "The Genetic Diversity of Mink (Neovison vison) Populations in China"

_animals, 2023, doi:10.3390/ani13091497_

Round 1

Reviewer 1 Report

The manuscript compares the genetic variability of different colour forms of mink, the topic is processed by modern NGS methodology. I have several questions and comments on the paper.

Line 110: It would be useful to add for which colour variant the reference genome is available (presumably for white?).

Lines 87-90, 281: Many stylistic errors in the colour names (e.g. "Sliverblue" mink)- probably due to automatic corrections

Lines 148-153: Very unclear text describing SNP counts- line 148 gives a total of 1,396,257 high-quality SNPs, but line 150 gives SNP counts for each colour variant ranging from 9,211,330 to 38,687,586- much more, so probably given for all individuals within that variant. However, this is not clear from the text.

Table 2: It would be useful to add a row about how many SNPs there are in total if we take all 96 individuals as one group. In addition to the row characterizing of each population, it would be useful to mention the number of individuals in that population (as this may affect the number of polymorphisms detected). It would also be helpful to indicate how many SNPs were unique to a given colour variant.

Line 188: It would be useful to mention whether there was a difference between some pairs of populations according to Fst (p would account for the different numbers of animals of each colour variant).

Discussion: If known, it would be useful to clearly mention (or insert a diagram) the phylogeny of the origin of each colour variant and which colour variant arose from which (at least for the colours analysed). It would also be mention which studies are done on neutral (e.g. microsatellite) markers or on markers potentially under selection.

Lines 259-260: It should be stated that these citations are also results of mink analyses

Author Response

Dear editors and reviewers,

Thank you very much for your critical review on our work. This is the list of corrections of our manuscript entitled “The genetic diversity of mink (Neovison vison) populations in China" (2247948) submitted to the Animals. According to your suggestion, we have dealt with the comments of the reviewers as follows:

We have change all the places based on the comments. The whole manuscript has been carefully checked again. Please feel free to inform me if there are still some questions.

With best regards,

Sincerely yours,

Zongyue Liu

Reviewer(s)' Comments to Author:

Question: Line 110: It would be useful to add for which colour variant the reference genome is available (presumably for white?)

Answer: Basing on the advice, we have adding some information on that. In line 110 is “Then, high-quality reads were aligned to the latest reference genome of standard black mink (https://www. ncbi. nlm. nih. gov/ assembly/ GCF_ 02017 1115.1/) using BWA-mem v 0.7.12.” Please check it, Thank you!

Question: Lines 87-90, 281: Many stylistic errors in the colour names (e.g. "Sliverblue" mink)- probably due to automatic corrections

Answer: Basing on the advice, we have changed “Sliver blue” to "Sliverblue". Please check it, Thank you!

Question: Lines 148-153: Very unclear text describing SNP counts- line 148 gives a total of 1,396,257 high-quality SNPs, but line 150 gives SNP counts for each colour variant ranging from 9,211,330 to 38,687,586- much more, so probably given for all individuals within that variant. However, this is not clear from the text.

Answer: The SNPs number of each animal are almost same, but the animal number of different color are total different, such as white type are 50, black type are 20, sliverblue are 10, brown type are 10 and red type are 7. So the Sum o SNPs for each colour group have great different, and varied from 9,211,330 to 38,687,586.

Question: Table 2: It would be useful to add a row about how many SNPs there are in total if we take all 96 individuals as one group. In addition to the row characterizing of each population, it would be useful to mention the number of individuals in that population (as this may affect the number of polymorphisms detected). It would also be helpful to indicate how many SNPs were unique to a given colour variant.

Answer: Basing on the advice, we have adding a row in the bottle of the Talbe 2. Please check it, Thank you!

Question: Line 188: It would be useful to mention whether there was a difference between some pairs of populations according to Fst (p would account for the different numbers of animals of each colour variant).

Answer: There is no P vaule in the calculation of Fst between each pairs of populations, but we can be analyzed based on the Fst. For example: Fst=0~0.05: the differentiation was small and could not be considered. Fst=0.05~0.15: there was a moderate degree of genetic differentiation among populations. Fst=0.15~0.25: high genetic differentiation among populations; Fst > 0.25: great genetic differentiation between populations. Note:The Fst value is between 0.15 and 0.25, with significant difference. The Fst value >0.25, and there is a great significant difference.

Question: Discussion: If known, it would be useful to clearly mention (or insert a diagram) the phylogeny of the origin of each colour variant and which colour variant arose from which (at least for the colours analysed). It would also be mention which studies are done on neutral (e.g. microsatellite) markers or on markers potentially under selection.

Answer: Basing on the advice, we have adding a Table 6 about coat color variant. Please check it, Thank you!

Question: Lines 259-260: It should be stated that these citations are also results of mink analyses.

Answer: we have changed that “This number of SNPs identified was significantly higher than that reported previously; for instance, 52,714 SNPs of the LD pattern across the American mink genome by GBS [25] and 34,816 SNPs  with body size and pelt length in American mink by GBS [15]. Please see line 259-260.Thank you!

Reviewer 2 Report

The author investigate the genetic diversity of mink populations in China and identify some genes associated with traits using the ddRAD-seq technology. This is an interestingly work. However, the description of Method section is poor, which prevent the reader from judging whether the findings of the study are reliable. The following concerns are need to be solved:

1. Line 111, the SNP calling is very important for readers to understand the reliable of ddRAD-seq data. Please provide the detail information on how to conduct the SNP calling using the GATK4.

2.Line 115, please provide the parameters involved in the SNP filtering.

3.Line129, I can’t look how the author perform the selective sweeps in the mink population

So, the above problems are not solved, I think the latter result is plausible

Minor comments

1. Line 30, Please fixed “1,3M”

2. Line 31, change “SNPs” to “SNP”

3. Line 36, please fixed this sentence.

4. Line87-92, please rewrite it

Author Response

Reviewer 2

Question: Line 111, the SNP calling is very important for readers to understand the reliable of ddRAD-seq data. Please provide the detail information on how to conduct the SNP calling using the GATK4.

Answer: The BWA-MEM (0.7.17-r1188) program was then used to compare the high-quality data against the reference genome  using default parameters. samtools 1.8 software was used to sort the SAM files and convert them into BAM files. Reads near InDels are most prone to mapping errors. To minimize the identification of SNPs caused by mapping errors, the reads near InDels were re-compared to improve the accuracy of the SNP calling. The Indel realigner command in the GATK 3.8 program was used to re-compare all reads near InDels to improve the accuracy of SNP prediction.

Question: Line 115, please provide the parameters involved in the SNP filtering.

Answer: we have added information in line 115. (1) Fisher test of strand bias (FS) ≤ 60;(2) HaplotypeScore ≤ 13.0;(3) Mapping Quality (MQ)≥ 40;(4) Quality Depth(QD) ≥ 2;(5) ReadPosRankSum≥ -8.0;(6) MQRankSum> -12.5;Please check it, Thank you!

Question: Line129, I can’t look how the author perform the selective sweeps in the mink population.

Answer: To detect regions with significant signatures of selective sweep, we considered the distribution of the π ratios and Fst values. We used an empirical procedure and selected windows simultaneously with significant low and high π ratios (the 5% left and right tails) and significant high Fst values (the 5% right tail) of the empirical distribution as regions with strong selective sweep signals along the genome, which should harbour genes that underwent a selective sweep.

Question: Line 30, Please fixed “1,3M”

Answer: We have changed “1,3M”to “1.3 million” in line29. Please check it, Thank you!

Question: Line 31, change “SNPs” to “SNP”

Answer: we have changed “SNPs” to “SNP”. Please check it, Thank you!

Question: Line 36, please fixed this sentence.

Answer: we have rewritten this sentence to “Two populations with different coat colors were subjected to the selection signature analysis, and 2300 genes in all were found to have a clear selection signature.” in line33. Please check it, Thank you!

Question: Line87-92, please rewrite it

Answer: we have changed to “A total of 97 unrelated minks were obtained from three different farms in China and included 50 white minks (WM), 20 black minks (BLM), 10 sliverblue minks (SM), 10 brown minks (BM), and 7 red minks (RM). WM, SM, and BM were from Huilongwan breeding mink farm in Yichun City; BLM was from Zhuojia mink farm in Jinlin City; and RM was from Shandong mink farm in Weifang City.” in line86-88. Please check them, Thank you!

Reviewer 3 Report

General comments:

This article is based on the genetic diversity of mink populations bred in China and presents new insights of 5 different Chinese mink colors varieties referring to genetic diversity, structure and selection signatures. Very interesting study about mink population supplying new information for further genomic studies with a deeper knowledge on important SNP variations. Overall with good methodology and stimulating results.

Dear authors please pay attention to several comments done during the text referring to plural vs singular accordance and the construction of sentences, in terms of grammatical issues and verbs. Attention to the text format with several spacing lacking near the references. Quality of figures must be improved and increased as several of them we cannot read to be able to highlight the overall quality of the manuscript.

References must also be revised as they are incomplete.

Good job!

 Specific comments:

L17 – don´t understand what you mean with “… is feeding across North zone of China….” Please clarify.

L21 – first time you mention GO and KEGG should be written what it means and you only do that after in abstract. Change to here.

L27 – “and now are wide-spread…” Consider changing the verb to “is wide spread”? Singular not plural?

L43 – use keywords different from title of article to increase possibility of searching and finding this article.

L48-49 - rephrase sentence, clarify.

L52-55 – reference needed to support this text

L59 – “… [4], 20….populations [5]” Space needed before 20 and after populations.

L60 –“….types [6] and…” attention to text format, with spacing needed between references. Revise in entire article.

L77 – “…SNPs were extracted…” Verb missing? Consider revision

L78 – scanning genome was used …?

L87-L99 – different text format and spacing. Homogenize in entire document.

L88 – 20 black minks - plural? Like the others?

L89-90 revise plural vs singular for mink!

L92 – blood sample were or blood samples? Revise accordance of plural and singular in all document

L118 . avoid using “WE used….”. Better to write “Plink was used …..”. Again in L119-120

L127-128 – something missing in this last sentence. Clarify/Rephrase

L132 – something missing in this sentence. For what? Clarify

Table 1 and 2 and following ones– a description for populations WM, BLM… needed in the end of each table

L165-166 spacing needed before table 3. Equal to the following ones.

Table 3 – indicate the meaning of abbreviations such as SNV bellow

Figure 1 A and B on same spot! Really like this? Not one in each graph?

A and B legends and titles of XX and YY– not readable, to small. Increase size

Table 5 – remove 1st column. Doing nothing there

Figure 2A – too small and we can´t understand anything. Also same comments as for the previous figure. Increase text size on axes and legend (A,B,C).

Figure 2 C – could try to use color for the animals on structure similar to Mink color populations to help readers to understand and realize differences.

Figure 3A-B-C unreadable! Increase size, mainly for B!

L281 - Do not start a sentence by …”And….”

L334 - … we found…. Rather than we find?

L337-338 -sentence not clear. Rephrase

L373 – “… for revising the article.”

L375 – revise all references as they are incomplete and not referred in the correct sequence. See journal guidelines to change accordingly. Lacking journals volumes, editions and page numbers.

Author Response

Reviewer 3

Question:L17 – don´t understand what you mean with “… is feeding across North zone of China….” Please clarify.

Answer: We have rewritten this sentence in line16, Please check it, Thank you!

Question:L21 – first time you mention GO and KEGG should be written what it means and you only do that after in abstract. Change to here.

Answer:We have changed “By GO and KEGG” to “Gene ontology and the Kyoto Encyclopedia of Genes and Genomes analysis showed that ” in line 20. Please check it, Thank you!

Question :L27 – “and now are wide-spread…” Consider changing the verb to “is wide spread”? Singular not plural?

Answer : We have changed “and now are wide-spread…” to “and is now widespread…”in line 25. Please check it, Thank you!

Question: L43 – use keywords different from title of article to increase possibility of searching and finding this article.

Answer: We have added “dd-RAD” in the Keywords.

Question: L48-49 - rephrase sentence, clarify.

Answer: We have rewritten the sentence in line46-48. Please check it, Thank you!

Question:L52-55 – reference needed to support this text

Answer: We have add 2 reference to support this text.“Although China lack a local mink, many breeds of minks were  imported from Russia, North America, and North Europe [2]. In an aim to avoid decrease in the number of purebred mink performances, the crossing method was used to foster the new type in most of the mink farms. This practice has led to the official recognition of 13 mink breeds , of which seven and six are domestically fostered and introduced breeds, respectively[4].”Please check it.

Question: L59 – “… [4], 20….populations [5]” Space needed before 20 and after populations.

Answer: We have added some space before 20 and after populations in “… [4], 20….populations [5]” . Please check them in line 57, Thank you!

Question: L60 –“….types [6] and…” attention to text format, with spacing needed between references. Revise in entire article.

Answer: We have adjusted the citation format in entire article. Please check them, Thank you!

Question:L77 – “…SNPs were extracted…” Verb missing? Consider revision

Answer:We have added the verb “ were” in the sentence. Please check it in line 77, Thank you!

Question:L78 – scanning genome was used …?

Answer:We have changed the sentence to “Scanning of the genome revealed putative genomic regions in American mink responsible for the response during Aleutian disease...” in line78. Please check it, Thank you!

Question:L87-L99 – different text format and spacing. Homogenize in entire document.

Answer: We have changed the format of them to make them consistent with the entire article. Please check it, Thank you!

Question:L88 – 20 black minks - plural? Like the others?

Answer: We have changed “20 black mink” to “20 black minks” in line87. Please check it, Thank you!

Question:L89-90 revise plural vs singular for mink!

Answer: We have changed the singular and plural of the mink in line86-88. Please check them, Thank you!

Question:L92 – blood sample were or blood samples? Revise accordance of plural and singular in all document

Answer: We have changed “blood sample ” to “blood samples ” in line90. Please check it , Thank you!

Question:L118 . avoid using “WE used….”. Better to write “Plink was used …..”. Again in L119-120

Answer: We changed “We used ...” to “ Plink was used ...” in line 127, and the same modification is reflected in line 128-129. Please check them, Thank you!

Question:L127-128 – something missing in this last sentence. Clarify/Rephrase

Answer: We have changed the last sentence to “, whereas the genetic analysis of population structure was carried out using ADMIXTURE software.”in line 136-137. Please check it, Thank you!

Question:L132 – something missing in this sentence. For what? Clarify

Answer: We have changed the sentence to “Furthermore, the KEGG enrichment analysis was performed using the KOBAS software and the KEGG website.”

Question: Table 1 and 2 and following ones– a description for populations WM, BLM… needed in the end of each table

Answer: We have added the description of the WM,BLM and others bellow the Table 1 and 2. Please check them, Thank you!

Question:L165-166 spacing needed before table 3. Equal to the following ones.

Answer:We have added some space before table 3 in line167. Please check it, Thank you!

Question:Table 3 – indicate the meaning of abbreviations such as SNV bellow

Answer: We have added the meaning of SNV and UTR bellow the Table 3. Please check them IN LINE180, Thank you!

Question:Figure 1 A and B on same spot! Really like this? Not one in each graph?A and B legends and titles of XX and YY– not readable, to small. Increase size

Answer:We have changed the position of the Figure 1 A,B and C and increased the size. Please check it, Thank you!

Question:Table 5 – remove 1st column. Doing nothing there

Answer: We have removed 1st column in Table 5. Please check it, Thank you!

Question:Figure 2A – too small and we can´t understand anything. Also same comments as for the previous figure. Increase text size on axes and legend (A,B,C).

Answer:We have changed the position of the Figure 2 A , B and C and increased the size. Please check it, Thank you!

Question:Figure 2 C – could try to use color for the animals on structure similar to Mink color populations to help readers to understand and realize differences.

Answer: In the original figure, we used 5 colors in the population genetic structure. Please check it, Thanks you!

Question:Figure 3A-B-C unreadable! Increase size, mainly for B!

Answer:We have changed the position of the Figure 3 A , B and C and increased the size. Please check it, Thank you!

Question:L281 - Do not start a sentence by …”And….”

Answer: We have changed the sentence to “ The first mutant of the regular mink, the Sliverblue mink, first appeared in Wisconsin, USA, in 1931.”.in line382-383. Please check it, Thank you!

Question:L334 - … we found…. Rathe than we find?

Answer:We have changed “we found” to “we find” in line 436. Please check it, Thank you!

Question:L337-338 -sentence not clear. Rephrase

Answer:  We have changed the sentence to “Many cell growth factors have been reported to be regulators of hair follicle growth and development, including hepatocyte growth factor (HGF), insulin-like growth factor-1 (IGF-1),and epidermal growth factor (EGF).” Please check it, Thank you!

Question:L373 – “… for revising the article.”

Answer: We have changed “Haihua Zhang for the revising the article” to “Haihua Zhang for revising the article” in line 482. Please check it, Thank you!

Question:L375 – revise all references as they are incomplete and not referred in the correct sequence. See journal guidelines to change accordingly. Lacking journals volumes, editions and page numbers.

Answer: We have changed all referenecs.

Round 2

Reviewer 2 Report

The revised manuscript has been greatly improved according to the comments. However, one question needs to be solved. Line 135-142, these sentences just described  the function annotation of candidate genes.  So, the author should changed these sentences before accepting. 

Author Response

Dear editors and reviewers,

Thank you very much for your critical review on our work. This is the list of corrections of our manuscript entitled “The genetic diversity of mink (Neovison vison) populations in China" (2247948) submitted to the Animals. According to your suggestion, we have dealt with the comments of the reviewers as follows:

We have change all the places based on the comments. The whole manuscript has been carefully checked again. Please feel free to inform me if there are still some questions.

With best regards,

Sincerely yours,

Zongyue Liu

Reviewer(s)' Comments to Author:

Reviewer 2 in Round 2

Question: The revised manuscript has been greatly improved according to the comments. However, one question needs to be solved. Line 135-142, these sentences just described  the function annotation of candidate genes.  So, the author should changed these sentences before accepting.

Answer: we have changed to “Candidate genes in the sweep regions of different mink populations were analyzed using the Gene Ontology (GO) and Kyoto Encyclopedia of Genes and Genomes (KEGG) databases. TopGO software was used to perform the GO enrichment analysis and KOBAS software and the KEGG website were used for the KEGG enrichment analysis. Calculated p-values were corrected for the false discovery rate (FDR) while applying an FDR threshold of ≤0.05. Pathways meeting the criteria were considered significantly enriched pathways. Please check it. Thanks.
